# Extreme haplotype variation in the desiccation-tolerant clubmoss *Selaginella lepidophylla*

Robert VanBuren [1,2], Ching Man Wai[1], Shujun Ou [1,3], Jeremy Pardo [4], Doug Bryant[5], Ning Jiang[1,3], Todd C. Mockler [5], Patrick Edger[1,3] & Todd P. Michael [6]

Plant genome size varies by four orders of magnitude, and most of this variation stems from dynamic changes in repetitive DNA content. Here we report the small 109 Mb genome of *Selaginella lepidophylla*, a clubmoss with extreme desiccation tolerance. Single-molecule sequencing enables accurate haplotype assembly of a single heterozygous *S. lepidophylla* plant, revealing extensive structural variation. We observe numerous haplotype-specific deletions consisting of largely repetitive and heavily methylated sequences, with enrichment in young *Gypsy* LTR retrotransposons. Such elements are active but rapidly deleted, suggesting "bloat and purge" to maintain a small genome size. Unlike all other land plant lineages, *Selaginella* has no evidence of a whole-genome duplication event in its evolutionary history, but instead shows unique tandem gene duplication patterns reflecting adaptation to extreme drying. Gene expression changes during desiccation in *S. lepidophylla* mirror patterns observed across angiosperm resurrection plants.

[1] Department of Horticulture, Michigan State University, East Lansing, MI 48824, USA. [2] Plant Resilience Institute, Michigan State University, East Lansing, MI 48824, USA. [3] Ecology, Evolutionary Biology and Behavior, Michigan State University, East Lansing, MI 48824, USA. [4] Department of Plant Biology, Michigan State University, East Lansing, MI 48824, USA. [5] Donald Danforth Plant Science Center, St. Louis, MO 63132, USA. [6] J. Craig Venter Institute, La Jolla, CA 92037, USA. Correspondence and requests for materials should be addressed to R.V. (email: bobvanburen@gmail.com) or to T.P.M. (email: toddpmichael@gmail.com)

The ability to survive periodic drying was essential for the aquatic ancestors of terrestrial plants during their transition from sea to land. These early protective mechanisms are thought to be ancestral and conserved in most land plants[1], forming the basis of modern seed and pollen desiccation pathways[2]. A small group of plants have maintained this ancestral desiccation tolerance, surviving near complete anhydrobiosis (<10% relative water content) during prolonged drought events. Desiccation tolerance is widespread in non-seed plants such as mosses, ferns, and lycophytes[3, 4], but is uncommon in angiosperms. High-quality genomes for several angiosperm resurrection plants have shed light on the molecular basis of desiccation tolerance[2, 5, 6], but the sparse genomic resources for non-seed plants has hindered progress toward understanding the origin of this trait.

Here we report the small genome of *Selaginella lepidophylla*, a lycophyte that can survive the loss of >95% cellular water for decades in a quiescent state. *Selaginella* are among the earliest diverging lineages of vascular plants, and extant species bear striking morphological similarities to fossils from the upper Carboniferous (333–350 Myr ago)[7, 8]. Genome size variation within this ancient group is minimal and *Selagienlla* are the only clade of land plants with no shared whole-genome duplication event[9]. Detailed comparative genomics across angiosperms and with the *S. moellendorffii* genome[10] shed light the origin of desiccation tolerance and unique properties of *Selaginella*.

## Results

**Haplotype-level genome assembly and annotation.** Extensive within genome heterozygosity exists for most plant species, and this variation enhances fitness and serves as a reservoir for adaptive evolution. Heterozygosity is a major issue in plant genome assembly, and most heterozygous plant genomes are highly fragmented with collapsed haplotype regions[11]. *S. lepidophylla* has an estimated genome size of 109 Mb based on kmer frequency analyses, which is consistent with the small genomes observed across the genus[9]. Despite its compact size, the *S. lepidophylla* genome is highly heterozygous, with a bimodal distribution of kmers corresponding to distinct heterozygous and collapsed homozygous regions (Supplementary Fig. 1). Initial sequencing attempts using an Illumina only approach yielded a highly collapsed and fragmented assembly of 21,282 scaffolds, totaling 54 Mb, with an N50 of 3 kb (Supplementary Table 1). To overcome assembly issues related to heterozygosity, we generated 175× coverage (19.3 Gb) of single-molecule real-time (SMRT) sequences from PacBio with a filtered subread N50 length of 18 kb (Supplementary Fig. 2). The long read lengths of PacBio sequencing facilitate accurate separation and assembly of heterozygotic haplotypes[12]. Raw PacBio reads were error corrected and assembled using the Canu assembler[13], which utilizes optimized overlapping and assembly algorithms to avoid collapsing repetitive regions and haplotypes. The resulting contigs were polished using the raw PacBio data with Quiver[14], followed by a second round of polishing with Pilon[15] using the high-coverage Illumina data.

The final *S. lepidophylla* assembly spans 122 Mb across 1149 contigs with an N50 length of 163 kb (Supplementary Table 2). These assembly metrics are similar to the *S. moellendorffii* genome (contig N50 length = 120 kb), which was sequenced using a reiterative Sanger approach[10]. Our assembly includes 15 of the 20 terminal telomere tracks ($2n = 20$) and 128 rRNA arrays,

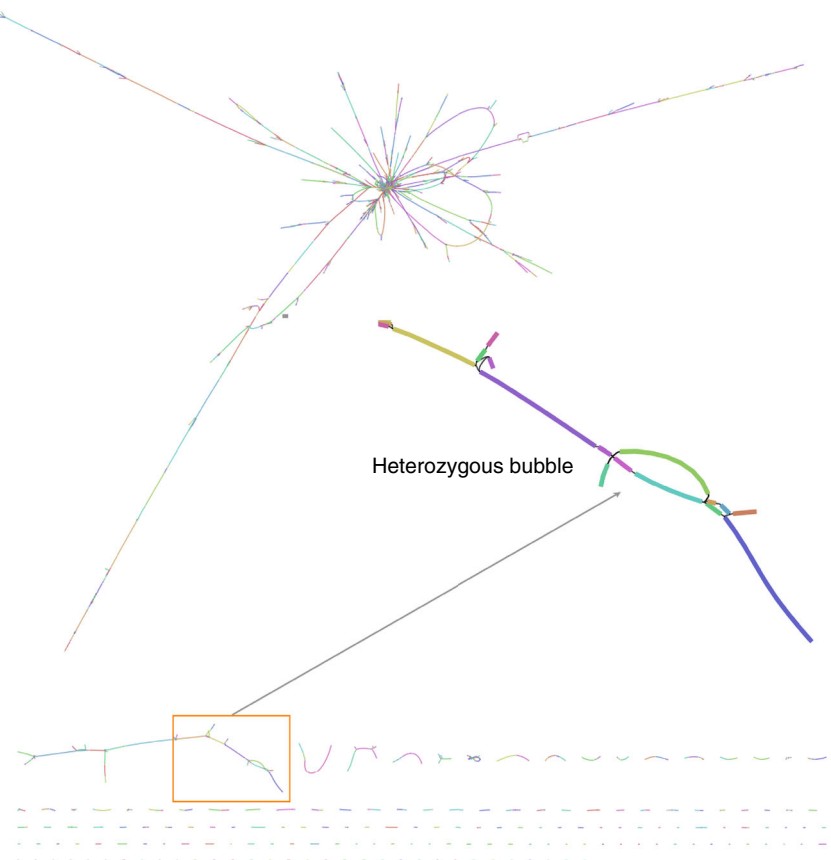

**Fig. 1** Genome assembly graph of *S. lepidophylla*. Each line (node) represents a contig with connections (edges) representing ambiguities in the graph structure. A subset of the graph showing heterozygous bubbles is enlarged. Contig color is randomly assigned

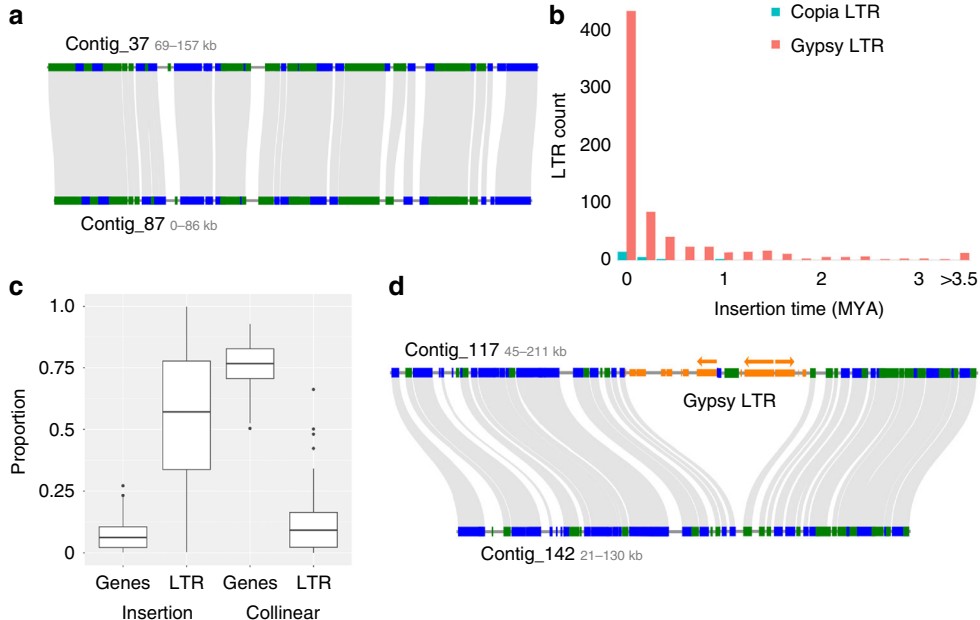

**Fig. 2** Extensive haplotype-specific LTR retrotransposon accumulation and deletion. **a** Typical micro-collinearity between two *S. lepidophylla* haplotypes. **b** Estimated insertions times of intact *Copia* and *Gypsy* LTR retrotransposons. **c** Composition of haplotype specific and paired collinear regions. The proportion of genes and LTRs in the 105 manually curated insertions (left) and flanking collinear regions (right) are plotted. **d** Micro-collinearity between two haplotypes showing a large 57 kb deletion in contig 142. Green and blue bars delineate gene orientation. Predicted LTR elements are depicted in orange and the three complete Gypsy LTRs are denoted by arrows. The three complete Gypsy LTRs have insertion times of <0.1 Ma

which are typically unassembled in most next-generation sequencing (NGS) based genomes[11]. The *S. lepidophylla* assembly graph is highly complex with numerous interconnected edges, reflecting bubbles and ambiguities caused by heterozygous regions (Fig. 1). The long PacBio reads assembled 930 haplotype blocks spanning 33.7 Mb of separated haplotype sequences (Methods section; Fig. 2a; Supplementary Fig. 3). Other haplotype aware assemblers were tested but Canu provided the best separation of haplotype blocks as well as the best contiguity.

We annotated 27,204 genes across a deduced 89 Mb haploid assembly, considerably more than *S. moellendorffii* (22,285 genes)[10]. The majority (96%) of core eukaryotic BUSCOs[16] were identified in the gene set, supporting the completeness of the genome assembly and quality of the annotation (Supplementary Table 3). Roughly 45% of genes (11,847) are found in duplicated pairs across the full 122 Mb assembly with an average protein homology of 99.1%. Gene density in *S. lepidophylla* is 4.0 kb per gene, which is higher than *S. moellendorffii*[10] and Arabidopsis[17], but lower than the smallest sequenced plant genome (82 Mb) of *Utricularia gibba*[18].

**Genome evolution and haplotype-specific structural variation.** Genome size varies by four orders of magnitude across land plants, ranging from 60 Mb in the carnivorous *Genlisea* to 152,000 Mb in *Paris japonica*. Lycophytes have a narrow variation in genome size compared to angiosperms[19], and *Selaginella* sizes range from 90 to 182 Mb[9]. Rapid changes in genome size are driven by bursts of long terminal repeat (LTR) retrotransposon proliferation followed by aggressive deletion through unequal homologous recombination and double-strand break repair[20]. LTR elements are distributed non-randomly across the genome with *Gypsy*-like elements predominantly found in gene poor and repeat rich blocks while *Copia*-like elements are dispersed among genes (Supplementary Table 4). Most of the LTR elements are clustered in a relatively small number of LTR families (Supplementary Figure 4). We identified 744 full-length LTR

retrotransposons (LTR-RTs), including 572 *Gypsy* and 17 *Copia* elements (Supplementary Table 5). The majority of intact LTRs are relatively young, with average insertion times of 0.1 and 0.4 MYA for *Copia* and *Gypsy* elements, respectively, and most intact LTR elements are younger than 0.5 Ma (Fig. 2b and Supplementary Fig. 5). The proportion of young LTR elements is higher in *S. lepidophylla* than other high-quality plant genomes (Supplementary Figure 5). The recent LTR amplification suggests most elements have been active recently in *Selaginella* but they are rapidly fragmented and deleted. *S. lepidophylla* has a much higher ratio (4.06, Wilcoxon rank-sum, $P < 0.05$, same statistical test hereafter unless specified) of solo-intact LTR elements comparing to the genomes of rice and Arabidopsis (2.71 and 1.80, respectively) (Supplementary Fig. 6), indicating more effective removal of LTR-RTs via unequal homologous recombination. Moreover, the average age of LTR families in *S. lepidophylla* is much younger (0.37 MY vs. 0.92 and 0.94 MY, respectively; $P < 0.05$) under the mutation rate of $\mu = 1.3 \times 10^{-8}$ (per bp per year), suggesting rapid removal of LTR elements. The intensive and rapid removal of LTR-RTs is likely one of the causes of haplotype variation and enrichment of LTR-RTs (Fig. 2).

Haplotype blocks within the *S. lepidophylla* genome are largely collinear, with conserved gene content and 98.2% sequence homology (Fig. 2a; Supplementary Fig. 3). However, since our analysis is based on both the number of contiguous protein and sequence homology, it is possible that we have missed large highly diverged haplotype blocks. A detailed comparison uncovered 104 large-scale haplotype-specific insertions or deletions (indels) across the genome. These high-confidence indels collectively span 2.5 Mb among 13.5 Mb of collinear sequences (18.5%) with individual sizes ranging from 4.8 kb to 85 kb with an average size of 24 kb (Supplementary Table 6). This is likely an underestimation as separate haplotypes were assembled for 35% of the genome and only high-confidence collinear regions spanning >75 kb were surveyed. Haplotype-specific regions are largely repetitive with 58% of bases consisting of LTR retrotransposons and only 9% genes (Fig. 2c). Flanking collinear regions show the

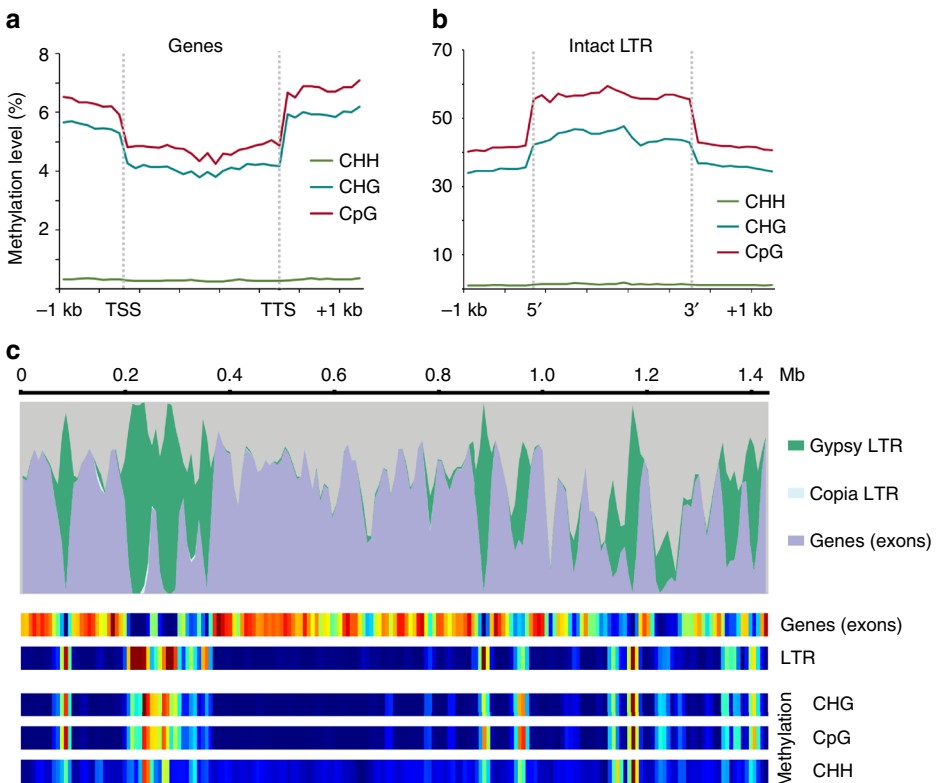

**Fig. 3** Global methylation patterns in the *S. lepidophylla* genome. **a** *S. lepidophylla* lacks gene body methylation. Levels of CHH (green), CHG (blue) and CpG (red) methylation are plotted in the upstream (transcriptional start site; TSS), downstream (transcriptional termination site (TTS), and body of genes. **b** Intact LTR retrotransposons are highly methylated with decreasing levels at the 5'- and 3'-flanking regions. **c** Methylation levels are highly variable across the genome with strong correlation of LTR retrotransposon density. Top, genome landscape of genomic features across Contig 1. Bottom, heatmap of genomic features and methylation levels, where blue indicates low abundance and red signifies high abundance

opposite trend, with 86% genic and only 11% LTR retro-transposon sequences. We annotated 335 protein-coding genes in the gene poor, haplotype-specific regions, but most (74%; 248 genes) have little to no detectable expression (TPM <0.5). Young LTR retrotransposons (insertion times <0.1 Ma) are over-represented in haplotype-specific regions, supporting a recent origin of these indels (Fig. 2d; $P < 0.01$).

DNA methylation is associated with suppressing repetitive element proliferation and larger, repeat-dense plant genomes tend to have higher levels of methylation[21]. We surveyed genome-wide DNA methylation in *S. lepidophylla* for patterns related to repetitive element silencing. Similar to other non-seed vascular plants[22, 23], *S. lepidophylla* lacks evidence of any gene body methylation (Fig. 3a). In contrast, intact LTR retro-transposons have high levels of CpG and CHG methylation with comparatively little CHH methylation (Fig. 3b). Methylation is highest in the body of LTR retrotransposons with levels dropping at the 5'- and 3'-flanking regions. Gene rich regions of the *S. lepidophylla* genome are highly compact with an average of 1.1 kb intergenic sequences with interspersed, repeat-dense blocks (Fig. 3c). This creates large tracks in the genome that are essentially devoid of methylation flanked by densely methylated, likely heterochromatic regions.

**Comparative genomics across *Selaginella* and angiosperms.** *Selaginella* arose in the early Carboniferous period 333–350 MYA[8], and the ancestors of *S. lepidophylla* and *S. moellendorffii* diverged about 250 MYA[9], well before the radiance of modern angiosperms[24]. Despite this separation, both lineages have maintained a similar genome size (~100 Mb) and neither has

experienced a recent whole-genome duplication event, contrast-ing patterns observed in all other land plants[10, 25]. *S. lepidophylla* and *S. moellendorffii* have maintained a surprising degree of collinearity with 36% (7849) of *S. moellendorffii* genes showing conserved order in *S. lepidophylla*. Genic regions are generally more compact in *S. lepidophylla*, correlating with the higher TE content (37.5%) in *S. moellendorffii* (Supplementary Fig. 7). By contrast, only ~15% of rice and core eudicot genes are colli-near[26], reflecting the role of WGD events in shaping gene order and genome architecture. In the absence of WGD, any new traits and adaptations associated with duplicate genes in *Selaginella* must arise through local or segmental gene duplications[27]. *S. moellendorffii* has significantly more tandem gene duplications (TD) than *S. lepidophylla* with 6693 compared to 4170, but both share similar enrichment pattern (Supplementary Tables 7–10). TDs in *Selaginella* are enriched in molecular function GO terms related to kinase activity, ion and other small molecule binding among others (Supplementary Tables 7 and 9). Few TDs were annotated as transcription factors or highly connected metabolic genes, which is consistent with expectations under the gene bal-ance hypothesis[28, 29]. Both *Selaginella* species have overlapping KEGG pathway enrichments across TDs including phenylpro-panoid, cutin, and secondary metabolite metabolism (Supple-mentary Tables 8 and 10). *S. lepidophylla* has TDs enriched in phagosome pathways, which may be related to controlled autophagy during desiccation[30].

Lycophytes are sister to all other vascular plants including angiosperms, the most diverse extant plant group on the planet. We surveyed changes in orthogroup composition across angios-perms and the two *Selaginella* species to identify patterns associated with WGD and life history traits. We identified 5891

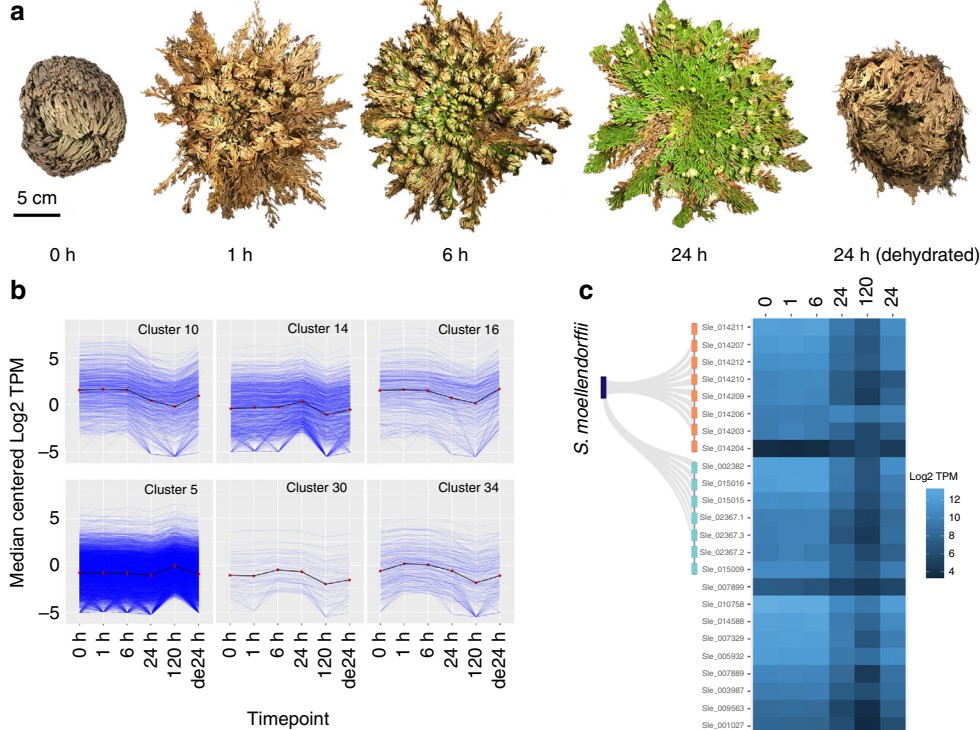

**Fig. 4** Genomic features of the rehydration and desiccation processes. **a** Overview of sampling for the rehydration and desiccation time course. Samples were taken from plants that were desiccated for 3 years (0), 1, 6, and 24 h post rehydration (recovery), 120 h post rehydration (fully recovered), and after a 24 h dehydration (de24 h). **b** Scaled transcript expression profiles (in transcripts per million, TPM) of representative gene co-expression clusters with decreased (10, 16, 34) and increased expression (5, 14, 34) during rehydration. **c** Heatmap of log2 transformed early light-induced protein (ELIP) expression with the two arrays of tandem duplicated genes in S. lepidophylla orthologous to the only ELIP in to S. moellendorffii highlighted on the left

orthogroups with representatives from every species and 69 orthogroups with expansion in angiosperms (Supplementary Table 11). Angiosperm-dominated orthogroups have diverse GO terms and enriched KEGG pathways for alkaloid biosynthesis, proteolysis, and plant-pathogen interactions (Supplementary Tables 12 and 13). The angiosperm biased orthogroups are more highly connected than expected by chance (~ 2.7 edges per gene; $P \leq 10^{-20}$, Fisher's exact test), fitting expectations for gene duplicates retained from WGD[28]. The 55 Selaginella-biased orthogroups have GO enrichment in vesicle related processes and KEGG pathway enrichment in phenylpropanoid biosynthesis (Supplementary Tables 14 and 15). Most Selaginella-biased orthogroups are unique and completely uncharacterized, reflecting the need for more functional studies in other major plant lineages.

**Origins and evolution of desiccation tolerance**. The transition from sea to land posed unique challenges for the ancestors of terrestrial plants. Early land plants needed to evolve protective mechanisms to survive periodic drying and low atmospheric water potential. Desiccation tolerance in angiosperms likely evolved from rewiring seed desiccation pathways[2], but the origins of this trait in non-seed plants is unknown. We surveyed gene expression changes during the desiccation and rehydration process in S. lepidophylla and compared patterns to angiosperm resurrection plants. S. lepidophylla can survive for years or even decades in a desiccated state without loss in viability. Samples were taken from S. lepidophylla plants that were desiccated (0 h), 1, 6, and 24 h post rehydration (recovery), 120 h post rehydration (fully recovered), and after a 24 h dehydration (Fig. 4a). Despite prolonged quiescence, 77% of genes (21,062) have detectable

expression in desiccated leaf tissue, implicating strong protective mechanisms to preserve the mRNA pool.

We performed gene-co-expression analysis across the rehydration and subsequent dehydration RNAseq timecourse data. The 38 co-expression clusters fall under the following three general patterns with (1) high expression in desiccated tissues followed by decreases during rehydration, (2) increased expression during rehydration, and (3) steady expression across all samples (Fig. 4b, Supplementary Fig. 8). Transcript levels are relatively consistent across the 0, 1, and 6 h post-rehydration time-points, with only 123 and 567 differentially expressed genes, respectively (0 h vs. 1 h and 1 h vs. 6 h). Clusters with increased expression during rehydration are enriched with functions related to DNA repair, oxidative and water stress response among others. Clusters with high expression in desiccated tissues and low expression during rehydration have functions related to cell protection and osmoprotectant accumulation as described in more detail below.

Gene duplications drive evolutionary innovation and tandem gene duplications (TDs) are important for adaptive evolution in dynamic environments[31]. Several clusters with peak expression during desiccation are enriched in TDs, including many with previously characterized roles in desiccation tolerance (Cluster 5, fdr = 1.26 $e^{-16}$; Supplementary Fig. 9). Early light-induced proteins (ELIPs) protect photosynthetic apparatus against photo-oxidative damage under high light and other abiotic stresses[32]. Land plants typically have 1–2 ELIPs but S. lepidophylla has undergone a major expansion to 23 ELIPs. Most ELIPs are located in three tandem arrays with 8, 7, and 2 members, and the largest array is syntenic to the single ELIP found in S. moellendorffii (Fig. 4c). ELIPs are universally highly expressed during desiccation and show co-expression with other desiccation-related genes including heat-shock proteins, DNA

repair enzymes, oleosins, LEAs, and others (enriched in cluster 10, fdr = 6.70 e$^{-23}$, Fig. 3c; Supplementary Table 16). A similar expansion of ELIPs was identified in the resurrection plant *Boea hygrometrica*[6], suggesting these proteins are likely involved in the convergent evolution of desiccation tolerance.

LEA proteins function in membrane stability and macro-molecular protection with roles in seed development, response to abiotic stresses, and desiccation tolerance in angiosperm resurrection plants[2, 33, 34]. *S. lepidophylla* has an expansion in LEA proteins compared to *S. moellendorffii* with 65 vs 36 across six LEA subfamilies (Supplementary Table 17). The LEA2 and seed maturation protein (SMP) subfamilies show the most dramatic expansion with 31 and 11 members, respectively. LEAs are dispersed across the genome and few are found in tandem arrays. LEAs are among the most highly expressed transcripts in the *S. lepidophylla* desiccation/rehydration timecourse with 48 showing significant expression changes (Supplementary Fig. 10). Interestingly, LEAs are also highly expressed in well-hydrated tissues, suggesting they may be involved in constitutive protective mechanisms, similar to several osmoprotectants.

Desiccation-related genes are found in large tandem arrays in the *S. lepidophylla* genome with individual genes having similar expression patterns. This reflects the importance of TD in the evolution of novel traits and the advantages of gene proximity for coordinated expression. Patterns of gene expression during desiccation in *S. lepidophylla* mirror the pathways activated in angiosperm resurrection plants. These pathways were either independently co-opted through convergent evolution in both lineages or present in the common ancestor of early land plants. The latter explanation is more likely as dehydration tolerance was acquired during the colonization of land. This hypothesis can be further refined with additional non-seed resurrection plant genomes.

The extreme haplotype-specific variation in *Selaginella* has not been observed in other sequenced plant genomes. This dynamic flux of accumulating and purging sequences likely serves two purposes. Strong selective pressures for removal of excess LTR retrotransposons act to preserve proper gene and *cis*-element spacing, which in turn maintains the small genome size. Unequal crossing over between divergent haplotypes and LTR-mediated gene duplications create a reservoir of new genes to facilitate adaptive evolution. In this context, haplotype-specific gene content may supplement the diversity lost in the absence of WGD.

## Methods

**DNA extraction and sequencing.** The *Selaginella lepidophylla* plants used in this study were collected in the Chihuahuan Desert in the Southwestern US. These plants were confirmed to be *S. lepidophylla* based on sequence homology to published vouchered specimens with the following nucleotide identities based on BLAST: rbcL (99.6%; GenBank: AF419051.1) ITS2 (99.8%; GenBank: AF419002.1) 28S (99.8%; GenBank: AJ507594.1). Desiccated plants were stored at 20 °C in dark conditions for roughly 3 years prior to this experiment. Rehydrated plants were maintained in growth chambers under 24 °C day, 20 °C night, 12-h photoperiod, and 200 µmol m$^{-2}$ s$^{-1}$ light. High-molecular-weight genomic DNA was extracted from young leaf tissue of a single *S. lepidophylla* plant using a modified nuclei preparation[35] followed by phenol chloroform purification to remove residual contaminants. DNA quality was verified on a low-concentration agarose gel (0.5% W/V) prior to library construction. gDNA was sheared to a 30 kb target size followed by end repair and adapter ligation. The final library was size selected using a BluePippen size selection system (Sage Science) followed by purification using AMPure XP beads (Beckman Coulter). High-molecular-weight libraries were sequenced using the PacBio RsII instrument with P6C4 chemistry. A total of 21 SMRT cells were sequenced. A PCR-free Illumina DNAseq library was constructed using the same high-molecular weight DNA following the manufactories instructions. Paired end 300 bp reads were sequenced on an Illumina MiSeq platform yielding a total of 80× Illumina sequence coverage for error correction.

**RNA extraction and sequencing.** *S. lepidophylla* plants were maintained in growth chambers under 16-h-light/8-h-night regime at 22 ± 2 °C for desiccation/

rehydration experiments. Sampling at the 0 h (desiccated), 24 and 120 h post rehydration and 24 h post dehydration were taken at *T* = 0 to avoid any variance associated with natural circadian oscillation. Sampling was done in triplicate with three biological replicates sequenced for each timepoint. The relative water content (RWC) for the *S. lepidophylla* rehydration time-course was calculated using the following equation: RWC = [(FW − DW)/(SW − DW)], where FW, DW, and SW indicate fresh weight, dry weight, and saturated weight, respectively. DW was obtained after drying plant leaf tissue at 103 °C for 24 h and SW was obtained after submerging the leaf tissue in water for 36 h. RWC for each sample can be found in Supplementary Table 18. Total RNA was extracted from 100 mg of ground *S. lepidophylla* leaf tissues using Omega-biotek E.Z.N.A. Plant RNA Kit (Omega-biotek), according to the manufacturer's instructions. Two micrograms of total RNA was used for construction Illumina TruSeq stranded mRNA libraries following the manufactures protocol (Illumina). Multiplexed pooled libraries were sequenced on the Illumina HiSeq4000 under paired-end 150 nt mode.

**DNA extraction and methylation sequencing.** Genomic DNA for methylation profiling was extracted from 500 mg of ground tissues using the CTAB method, followed by RNase digestion and column purification using Zymo Research genomic DNA clean & concentrator™ (Zymo Research). In total, 500 ng of genomic DNA was first sheared to an average size of 400–800 bp using Covaris M200. Libraries were constructed using KAPA Hyper Prep Kits (KapaBiosystem, #KK8504) according to the manufacturer's protocol. The sheared DNA fragments were end-repaired and A-tailed followed by adapter ligation. Adapter-ligated fragments were subjected to bisulfite conversion using EZ DNA Methylation-Lightning™ Kit (Zymo Research). Bisulfite-converted DNA was amplified using KAPA HiFi HotStart Uracil + Ready Mix (KapaBiosystem) with 8 PCR cycles. PCR products were then size-selected by gel electrophoresis for the range of 300–600 bp and recovered by Zymoclean® Gel DNA Recovery Kit (Zymo Research). Multi-plexed pooled libraries were sequenced on the Illumina HiSeq4000 platform under paired-end 150 nt mode.

**Genome assembly.** *S. lepidophylla* PacBio reads collectively span 19.3 Gb representing 175× genome coverage with a filtered subread N50 length of 18 kb. Raw reads were error corrected and assembled using the Canu (V1.4)[13] assembler with the following parameters: minReadLength = 3000, GenomeSize = 110 Mb, and minOverlapLength = 2000. Other Canu parameters were left as default. The Falcon assembler (V0.3.0) was also tested but the overall contiguity was much lower (N50–110 kb) and a higher proportion of haplotypes were collapsed. Contigs were polished using a reiterative approach with two rounds of Quiver (V2.3.0)[14] with a minimum subread length = 5000 bp, minimum polymerase read quality = 0.8, maximum divergence percentage = 30 and minimum anchor size = 15. The polished contigs underwent a final round of error correction using high-coverage 300 bp PE Illumina data using Pilon (V1.21)[15]. Quality trimmed Illumina reads were aligned to the Quiver polished contigs using bowtie2 (V2.3.0)[36] with default parameters. Reads were locally re-aligned around insertions/deletions (indels) using the IndelRealigner from the genome analysis tool kit (GATK; V3.7)[37]. Pilon parameters were as follows: --flank 7, --K 51, and --mindepth 30. Illumina reads were re-aligned to the final polished genome to verify no residual errors remained using GATK.

**Identification of haplotype-specific regions.** Assembled haplotypes were identified using a two-step approach based on gene level microsynteny followed by whole-genome alignment. Gene synteny avoided spurious hits and misclassification around repetitive elements with whole-genome alignment facilitating base-pair resolution of overlapping haplotype regions. Syntenic gene pairs were identified with BLAST followed by filtering for only collinear pairs using the MCSCAN toolkit (V1.1)[38]. This approach flagged 11,847 duplicated gene pairs collectively representing ~45% of genes across 28 Mb of the genome. The exact boundaries of haplotype overlaps and haplotype-specific structural variants were identified by whole-genome alignment using MUMmer (V3.0)[39]. The unmasked genome was aligned against itself using NUCmer with default parameters and a minimum overlap size of 2 kb. Only hits in contigs with collinear gene pairs were considered and overlaps in these regions were manually filtered removing misalignments in repetitive regions. In total, 33.7 Mb of assembled haplotype sequences were identified including 105 high-confidence haplotype-specific insertions/deletions (indel). The boundaries of each indel were manually refined using the GEvo module in CoGe (https://genomevolution.org)[40]. Haplotype-specific indels were verified through alignment of raw PacBio reads spanning the boundary regions in both haplotypes.

**Transposable element identification.** Non-autonomous DNA transposons in *S. lepidophylla* were identified by MITE-Hunter[41] and were classified according to the sequence of termini and length of target site duplication (TSD) provided in the output from MITE-Hunter based on the guide of Zhao et al.[42]. Candidate long terminal repeat retrotransposons (LTR-RTs) in *S. lepidophylla* were identified using LTR_Finder (V1.02)[43] and LTRharvest[44]. LTR_retriever[45] was used to filter out false LTR retrotransposons using structural and sequence features of target site duplications, terminal motifs, and LTR-RT Pfam domains. Whole-genome

annotation of LTR retrotransposons was done by RepeatMasker (http://www.repeatmasker.org/)[46] using the non-redundant LTR-RT library constructed by LTR_retriever. DNA transposons that were nest-inserted in LTR retrotransposons were identified and removed in the LTR library using a Perl scrip "Purger.pl" from the LTR_retriever package. Non-LTR retrotransposons and other interspersed repeats were identified using RepeatModeler (V1.0.8) (http://www.repeatmasker.org). While RepeatModeler also reports LTR retrotransposons, their sequences were examined using TE-specific Pfam domains by HMMER (V3.1b2)[47]. Among 34 RepeatModeler-predicted LTR retrotransposons, none of them were exclusively matched with LTR-specific Pfam domains, thus all these sequences were labeled as unknown repeats instead of LTR retrotransposons. Candidate tandem repeats were identified by Tandem Repeat Finder (trf409.linux64)[48] and were masked by the TEs identified in previous steps. Unmasked candidates were labeled as tandem repeats, and the sequence redundancy were reduced using Cd-hit (V4.6.6)[49] with parameters -c 0.8, -G 0.8, -s 0.9, -aL 0.9, -aS 0.9, -M 0, and -T 5. The Perl script "Purger.pl" from LTR_retriever and the plant protein database "alluniR-efprexp082813" from MAKER (V2)[50] were used to remove any protein-coding sequences that were accidentally identified in these TEs. The top 10 most abundant repeat families (based on genome mass) were manually curated through alignment of multiple family members so to determine their boundary, termini, and TSD, which were used to classify the relevant repeat.

Solo LTRs were identified from the whole-genome annotation of LTR-RT using the Perl script "solo_finder.pl" in the LTR_retriever package. In brief, independent LTR sequences (no other LTR-related sequence presented in 300 bp distance) that cover at least 80% of the library LTR entry with alignment score >300 and at least 100 bp in length were identified as solo LTR. The solo-intact LTR ratio was calculated based on the number of solo LTR over the number of intact LTR-RT for each family, which was estimated using the Perl script "solo_intact_ratio.pl" in the LTR_retriever package. Since LTR length is positively correlated with the formation of solo LTRs, we only compared the S/I ratio using elements with comparable LTR length. This was achieved by using the 95 percentile of LTR length in *S. lepidophylla* (90–1540 bp) to screen for LTR-RT families within into this range in the rice and Arabidopsis genomes.

Estimation of insertion time for each intact LTR element was provided by LTR_retriever using the formula of $T = K/2\mu$, where $K$ is the divergence rate approximated by percent identity and adjusted using the Jukes-Cantor model for non-coding sequences. The neutral mutation rate of $\mu = 1.3 \times 10^{-8}$ mutations per bp per year was used[51].

**Genome annotation**. The *S. lepidophylla* genome was annotated using MAKER (V 2.31.8)[52]. A set of high-quality transcripts for gene prediction were assembled with StringTie (V1.3.1)[53] using the stranded, paired end Illumina RNAseq data described above. Quality filtered RNAseq reads were aligned to the unmasked *S. lepidophylla* genome using TopHat (V2.1.0)[54] and the accepted_hits.bam file was used as input for StringTie. Default parameters were used for StringTie with the --merge flag to produce a set of non-redundant transcripts to feed into MAKER. These transcripts were treated as expressed sequence tag evidence and protein sequences from Arabidopsis[17], *Selaginella moellendorffii*[10], and UniprotKB plant databases[55] were used as protein homology evidence. Intact *S. lepidophylla* LTR retrotransposons were input as a custom repeat database along with default repeat libraries in MAKER. Ab initio gene prediction was done using the gene predictors SNAP and Augustus with reiterative training in *S. lepidophylla*. The initial MAKER Max gene set was filtered for gene models with Pfam domain evidence and annotation edit distance <1.0. Putative transposon-derived models were removed. This produced a set of 39,051 filtered gene models. A further 11,847 duplicated gene models corresponding to separate haplotypes were removed to produce a final set of 27,204 non-redundant genes.

**Orthogroup annotation**. Orthogroups were identified using Orthofinder (V1.0.6)[56] with protein-coding genes from the Arabidopsis[17], rice[57], poplar[58], grape[59], *Oropetium thomaeum*[60], *S. moellendorffii*[10], and *S. lepidophylla* genomes using default settings (Supplementary Table 11). We set a stringent cutoff of a 5 fold difference in orthogroup composition and at least a minimum of 10 genes (average) across either Angiosperms or *Selaginella* groups to identify expanded orthogroups. On average, angiosperm enriched orthogroups had 20-fold enrichment and *Selaginella* had 40 fold enrichment.

**Gene expression analysis**. Adapter sequences and low-quality bases were filtered from the raw, paired end Illumina RNAseq reads using TRIMMOMATIC (v0.33)[61] and reads shorter than 36 bp were discarded. Expression levels (in Transcript Per Million, TPM) were quantified using Kallisto[62] with the 27,204 monoploid *S. lepidophylla* gene models provided as input. An average TPM for each time-point was calculated from the mean TPM of the three biological replicates. We preformed weighted gene co-expression analysis of the dehydration and subsequent rehydration time course RNA-seq data with the R package WGCNA[63]. The expression data were pre-filtered to remove genes with excessive missing values or invariant expression using the built-in quality control function. A signed co-expression network was constructed using a soft-thresholding power of 10 and default parameters. The only exception was the mergeCutHeight parameter, controlling the minimum distance between co-expression clusters, which was set to

0.15. The co-expression network resulted in 38 clusters, with 13 genes not placed in any cluster.

**MethylC-seq analysis**. Quality filtered MethylC-seq reads were filtered as described above. Three replicates for 0 h, 24 h, and 120 h post rehydration and unconverted control DNA were sequenced. The average genome coverage of each library is 12.3×. Reads were mapped to the in silico converted, unmasked *S. lepidophylla* genome for the forward and reverse strands using Bismark (v0.12.0)[64]. Genome-wide cytosine methylation was obtained using the following parameters in the Bismark methylation extractor step: −comprehensive −CX_context --cytosine_report. The methylation level of each CHH, CHG, and CpG position was calculated as the number of methylated count/total of methylated and unmethylated count. Any position with coverage of less than three reads was removed. For each cytosine position, the average methylation level was calculated from the mean methylation level among biological replicates.

Methylation level of the upstream (1 kb), downstream (1 kb) and body (from transcription start site to transcription stop site) of all genic regions and intact LTR retrotransposons were calculated using BEDTools (v2.24.0)[65] and in-house scripts. To study differential methylated regions between 0 h, 24 h, and 120 h post rehydration samples, the whole-genome methylated count of each library was obtained from the Bismark output. Differential methylated regions were analyzed using the R package methylKit (v1.2.0)[66] in 100 bp window and minimum coverage of 10 reads per site.

**Data availability**. The genome assembly, raw PacBio, Illumina DNAseq, and RNAseq data are available from the National Center for Biotechnology Information (NCBI) Short Read Archive (SRA). The RNAseq reads were deposited to NCBI SRA under BioProject PRJNA420971 (BioSample SAMN08128719–08128734) and MethylCseq data were deposited under BioProject PRJNA421023 (BioSample SAMN08132846-08132853). The genome assembly has been deposited under BioSample SAMN07071123 and BioProject ID PRJNA386571.

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

## Acknowledgements

R.V. was supported by Hatch (Award 1013240) and startup funds from MSU. S.O and N. J. were supported by the National Science Foundation (MCB-1121650 to N.J.)

## Author contributions

R.V. designed and conceived research; T.P.M. performed Illumina and PacBio sequencing; R.V., J.W., S.O. and N.J. annotated genome features; J.W. and R.V. performed desiccation experiments, RNAseq, and MethylCseq experiments; R.V., J.W., S.O., J. P., D. B., T.C.M., P.E. and T.P.M analyzed data; R.V. wrote the paper. All authors read and approved the final manuscript.

## Additional information

**Competing interests:** The authors declare no competing financial interests.

