## [Peer Review File · Nature Communications]

PEER REVIEW FILE

Reviewers' comments:

Reviewer #1 (Remarks to the Author):

This is a highly professional study and product. The genome biology and comparative aspects of the study are not only first-rate but also state of the art. I commend the authors for a very thorough assembly and analysis. Although I will have to agree to disagree on a couple of the responses to the reviewers, the current manuscript has allayed the majority of the criticisms leveled at the earlier version.

The work is certainly original and important to the field and the major claims that the authors make are justified and defensible. I do think that it is premature to state that the majority of the variation in the genome of this single plant stems from dynamic changes in repetitive DNA content. Although the evidence is very suggestive that this is the case it does not warrant such a categorical statement. To conclude this would require a much more detailed phylogenetic and comparative genomic study and the sequencing of more than one individual (from an unknown habitat).

Although many of the aspects of the transcriptomic and annotation analyses are similar to what has been documented in the literature for over resurrection species the manuscript offers much and will certainly add to our understanding of the resurrection phenotype and stimulate discussion.

The statistical analysis, although I am not an expert, appear appropriate and generally accepted in the field.

Issues (minor and correctable).

Related to the question "comment on the the ability of a researcher to reproduce the work, given the level of detail provided". In this respect I would have to say that it would be very

difficult (even if the researcher had the resources and expertise to repeat the work). The genome sequence is from a single plant of unknown provenance (this should be added to the Materials and Methods section) and it is unclear if there is a voucher in place either for the material or the plant itself (I do not know if it can be propagated or cloned but tissue or the plant should be vouchered and this information should be provided). The dehydration rate for the transcriptomics profile was not reported and as dehydration rate is critical to the response this should be provided (as a drying curve) - this was not provided in the Supplemental Table 18. By the way - it is unclear what is meant by water content (%). Is this as described for seeds? - where $WC\% = \frac{\text{Fresh weight} - \text{dry weight}}{\text{fresh-weight}} \times 100$ - or is this simply a percentage of the original fresh weight (not ideal for comparative or repetition) - water content is very experiment specific so people usually report RWC (Relative Water Content) so that studies can be compared across and within species.

The comparative genomic aspects of the manuscript are somewhat limited (perhaps by necessity) but it would have been informative to include comparisons to the other resurrection plant genomes that have been published: *Xerophyta viscosa* and *Oropetium thomaeum*. A comparison with the two monocots and the dicot *Boea* would have been interesting.

Reviewer #2 (Remarks to the Author):

This revised manuscript by VanBuren et al is improved and I look forward to its publication. The authors have addressed my previous comments. However, the authors should have included in the manuscript the additional information that they provided as response to the reviewers, such as the provenance and identification of the specimens used in the study (reviewer #1) and the clarification about orthogroup expansions (reviewer #2). Including this information is important because readers might raise similar concerns. I also noticed that the authors still did not provide information about the storage conditions as asked by reviewer #1. Considering that enough information should be given as to allow other researchers to replicate the experiment, the storage conditions are crucial in case other researchers want to perform transcriptome analysis in dry stored tissues.

Reviewer #3 (Remarks to the Author):

The manuscript by VanBuren et al was clearly improved since I have seen it the last time, and many of my concerns have been addressed. But I still have two main points which I think should be addressed.

1) I still do not see why the haplotype differences are “extreme”. I would be surprised if this

level of difference (presence/absence variation of 104 LTRs) could not be found within other plant genomes as well. In fact, centromeric regions are full of LTRs in Arabidopsis and other plants, and are also highly variable. Why would those differences not come up in such other assemblies as well? Needless to say, the resolution of the assembly is excellent, and that such things can be found is great, but I do not see why the degree of difference should be special or extreme in any way.

2) The analysis of solo LTRs, showed that there is a low solo/intact LTR ratio, which implies a low rate of intra-element recombination or truncation of the full-length elements. The authors suggest that this does not result from a low deletion frequency in general, but that deletion/removal of LTR elements is driven by genome shuffling (line 98). I am not sure, why the authors can rule out low deletion rates. Moreover, the conclusion on genome shuffling is ignored in the rest of the manuscript, and rapid deletion of LTRs is suggested, but not justified. For example, in the abstract (Line 24: “but rapidly deleted, suggesting ‘Bloat and Purge’”), last paragraph (“dynamic flux of accumulating and purging sequences’) and other parts, the author insist on frequent deletions of the LTRs.

While the authors convinced me that the fraction of recent insertion is higher than in other genomes (fraction not amount), I do not find evidence for the high rate of deletion and the dynamics they mention.

Minor points

Line 116: Why would the concentration of LTRs explain why it is was possible to assemble such regions? Should not clusters of repeats be a reason for assembly breakage?

Responses to Reviewers:

Reviewers' comments:

Reviewer #1 (Remarks to the Author):

This is a highly professional study and product. The genome biology and comparative aspects of the study are not only first-rate but also state of the art. I commend the authors for a very thorough assembly and analysis. Although I will have to agree to disagree on a couple of the responses to the reviewers, the current manuscript has allayed the majority of the criticisms leveled at the earlier version.

The work is certainly original and important to the field and the major claims that the authors make are justified and defensible. I do think that it is premature to state that the majority of the variation in the genome of this single plant stems from dynamic changes in repetitive DNA content. Although the evidence is very suggestive that this is the case it does not warrant such a categorical statement. To conclude this would require a much more detailed phylogenetic and comparative genomic study and the sequencing of more than one individual (from an unknown habitat).

Although many of the aspects of the transcriptomic and annotation analyses are similar to what has been documented in the literature for over resurrection species the manuscript offers much and will certainly add to our understanding of the resurrection phenotype and stimulate discussion.

The statistical analysis, although I am not an expert, appear appropriate and generally accepted in the field.

Issues (minor and correctable).

Related to the question "comment on the the ability of a researcher to reproduce the work, given the level of detail provided". In this respect I would have to say that it would be very difficult (even if the researcher had the resources and expertise to repeat the work). The genome sequence is from a single plant of unknown provenance (this should be added to the Materials and Methods section) and it is unclear if there is a voucher in place either for the material or the plant itself (I do not know if it can be propagated or cloned but tissue or the plant should be vouchered and this information should be provided).

We have updated the methods to include a description of the plant material (see below and lines 225-240). To obtain enough HMW gDNA for PacBio sequencing and bisulfite-seq, we had to sacrifice the original plant used in this study. We did not anticipate such high heterozygosity and haplotype specific structural variation.

“The *Selaginella lepidophylla* plants used in this study were collected in the Chihuahuan Desert in the Southwestern US. These plants were confirmed to be *S. lepidophylla* based on sequence homology to published vouchered specimens with the following nucleotide identities based on

BLAST: rbcL (99.6%; GenBank: AF419051.1) ITS2 (99.8%; GenBank: AF419002.1) 28S (99.8%; GenBank: AJ507594.1). Desiccated plants were stored at 20C in dark conditions for roughly three years prior to this experiment. Rehydrated plants were maintained in growth chambers under 24C day, 20C night, 12-hour photoperiod, and 200 $\mu\text{molm}^{-2}\text{s}^{-1}$ light.”

The dehydration rate for the transcriptomics profile was not reported and as dehydration rate is critical to the response this should be provided (as a drying curve) - this was not provided in the Supplemental Table 18. By the way - it is unclear what is meant by water content (%). Is this as described for seeds? - where $\text{WC}\% = (\text{Fresh weight} - \text{dry weight}) / \text{fresh-weight} \times 100$ - or is this simply a percentage of the original fresh weight (not ideal for comparative or repetition) - water content is very experiment specific so people usually report RWC (Relative Water Content) so that studies can be compared across and within species.

We previously calculated the RWC (though erroneously labeled the table ‘Water Content’). We have added a description in the methods (see lines 250-253):

“The relative water content (RWC) for the *S. lepidophylla* rehydration time-course was calculated using the following equation: $\text{RWC} = [(\text{FW} - \text{DW}) / (\text{SW} - \text{DW})]$, where FW, DW and SW indicate fresh weight, dry weight, and saturated weight respectively. The DW was obtained after drying plant leaf tissue at 103 °C for 24 h and SW was obtained after submerging the leaf tissue in water for 36 h. RWC for each sample can be found in Supplemental Table 18.”

The comparative genomic aspects of the manuscript are somewhat limited (perhaps by necessity) but it would have been informative to include comparisons to the other resurrection plant genomes that have been published: *Xerophyta viscosa* and *Oropetium thomaeum*. A comparison with the two monocots and the dicot *Boea* would have been interesting.

We agree with the reviewer that additional comparative genomics analyses with other resurrection plants would be interesting, but this is beyond the scope of this manuscript.

Reviewer #2 (Remarks to the Author):

This revised manuscript by VanBuren et al is improved and I look forward to its publication. The authors have addressed my previous comments. However, the authors should have included in the manuscript the additional information that they provided as response to the reviewers, such as the provenance and identification of the specimens used in the study (reviewer #1) and the clarification about orthogroup expansions (reviewer #2). Including this information is important because readers might raise similar concerns. I also noticed that the authors still did not provide information about the storage conditions as asked by reviewer #1. Considering that enough

information should be given as to allow other researchers to replicate the experiment, the storage conditions are crucial in case other researchers want to perform transcriptome analysis in dry stored tissues.

We have included these details in the revision (See lines 228-231 and 364-372):

The *Selaginella lepidophylla* plants used in this study were collected in the Chihuahuan Desert in the Southwestern US. These plants were confirmed to be *S. lepidophylla* based on sequence homology to published vouchered specimens with the following nucleotide identities based on BLAST: *rbcL* (99.6%; GenBank: AF419051.1) *ITS2* (99.8%; GenBank: AF419002.1) *28S* (99.8%; GenBank: AJ507594.1). Desiccated plants were stored at 20C in dark conditions for roughly three years prior to this experiment. Rehydrated plants were maintained in growth chambers under 24C day, 20C night, 12-hour photoperiod, and 200 μ molm⁻²s⁻¹ light. High-molecular weight genomic DNA was extracted from young leaf tissue of a single *S. lepidophylla* plant using a modified nuclei preparation³⁴ followed by phenol chloroform purification to remove residual contaminants.

Orthogroup annotation

Orthogroups were identified using Orthofinder (V1.0.6) with protein coding genes from the *Arabidopsis*, rice, poplar, grape, *Oropetium thomaeum*, *S. moellendorffii*, and *S. lepidophylla* genomes using default settings (Supplemental Table 11). We set a stringent cutoff of a 5 fold difference in orthogroup composition and at least a minimum of 10 genes (average) across either Angiosperms or *Selaginella* groups to identify expanded orthogroups. On average, angiosperm enriched orthogroups had 20 fold enrichment and *Selaginella* had 40 fold enrichment.

Reviewer #3 (Remarks to the Author):

The manuscript by VanBuren et al was clearly improved since I have seen it the last time, and many of my concerns have been addressed. But I still have two main points which I think should be addressed.

1) I still do not see why the haplotype differences are “extreme”. I would be surprised if this level of difference (presence/absence variation of 104 LTRs) could not be found within other plant genomes as well. In fact, centromeric regions are full of LTRs in *Arabidopsis* and other plants, and are also highly variable. Why would those differences not come up in such other assemblies as well? Needless to say, the resolution of the assembly is excellent, and that such things can be found is great, but I do not see why the degree of difference should be special or extreme in any way.

We agree with the reviewer that haplotype differences are likely severely underestimated in most plant genomes. Haplotype-specific structural variations are probably present in most genomes, especially in centromeric regions (as the reviewer points out), and in genomes with high

heterozygosity. Extensive haplotype variation has not previously been identified on a genome-wide level and previous estimates are based on SNPs and small-scale Indels. Our work in *S. lepidophylla* is the first to report large, > 50kb structural variations that dwarf any previous estimates. Most of these structural variants are in gene dense regions, which typically show lower levels of polymorphisms, We feel ‘extreme’ is an apt description of these variants because of their sheer size and prevalence within a single plant. We speculate structural variants of this magnitude have not been described previously because of the following reasons:

1. Most high-quality plant genomes are from highly inbred lines that should have minimal to no structural variation within a single plant.
2. Most highly heterozygotic plant genomes (where we would expect structural variation) such as grape, pineapple, citrus, apple, etc. were assembled from doubled haploid lines or NGS technologies that failed to phase the haplotypes. Sequencing true diploid lines with high-coverage PacBio data might help resolve true haplotype differences.

2) The analysis of solo LTRs, showed that there is a low solo/intact LTR ratio, which implies a low rate of intra-element recombination or truncation of the full-length elements. The authors suggest that this does not result from a low deletion frequency in general, but that deletion/removal of LTR elements is driven by genome shuffling (line 98). I am not sure, why the authors can rule out low deletion rates. Moreover, the conclusion on genome shuffling is ignored in the rest of the manuscript, and rapid deletion of LTRs is suggested, but not justified. For example, in the abstract (Line 24: “but rapidly deleted, suggesting ‘Bloat and Purge’”), last paragraph (“dynamic flux of accumulating and purging sequences”) and other parts, the author insist on frequent deletions of the LTRs. While the authors convinced me that the fraction of recent insertion is higher than in other genomes (fraction not amount), I do not find evidence for the high rate of deletion and the dynamics they mention.

Thank you for the comments. We agree with the reviewer that the data presented in the original **Supplemental Figure 7** alone is not sufficient to justify the rapid LTR-RT removal in *S. lepidophylla*. We were actually puzzled about the apparent low intra-element recombination rate, inferred from the ratio between solo LTRs and intact elements ratio (called S/I ratio) in *S. lepidophylla*. During the revision, we carefully examined the element composition in four well-assembled genomes including *S. lepidophylla*, Arabidopsis, rice, and maize. The Arabidopsis genome and the rice genome are both small genomes which removal of LTR-RTs primarily through intra-element recombination as reported by Hu *et al.* (2011)¹ and Tian *et al.* (2009)². It turned out that *S. lepidophylla* not only has younger elements but also has elements with shorter LTRs. It is well known that LTR length is positively correlated with the formation of solo LTR, because larger LTR results more frequent alignment between the two LTRs of a single element (Du *et al.*, 2010)³. In addition, age is positively correlated with the formation of solo LTRs in that the longer time elapses, the more likely for intra-element recombination to occur. As a result, the apparent low SI could be an artifact of short LTRs and younger elements.

To test this hypothesis and remedy the LTR length effect on S/I ratio, we took the 95 percentile of LTR length from *S. lepidophylla* (90-1540 bp) and filtered for LTR families in both

Arabidopsis and rice that fell into this range, and the resulting LTR length distribution is largely comparable (see the mean LTR length in the new supplemental Figure 7) among the three species. We did intend to retrieve elements from maize within the same range; however, the mean LTR length ended up with > 1 kb, which is approximately twice as that for other genomes, so we did not include maize in the comparison. After filtering, the S/I ratio distribution of these three species were plotted as new **Supplemental Figure 7**. As we can see from the figure, under similar element compositions, the S/I ratio in Arabidopsis and rice is shifted to the lower end. As a result, the S/I ratio of *S. lepidophylla* is significantly higher than those in both rice and Arabidopsis (*t*-test, $P < 0.05$) (**Supplemental Figure 7**), indicating intensive deletion of LTR-RTs. Moreover, the average age of LTR families in *S. lepidophylla* is much younger, implying *S. lepidophylla* elements eliminated more elements using less time. This suggests rapid removal of LTR elements through intra-element recombination. We consider the original Supplemental Figure 7 was misleading and used this new plot to replace the original Supplemental Figure 7. “Genome shuffling” was originally used in micro-organism engineering, when genomes of different bacteria strains were merged and formed mosaic haplotypes. Although we observed rapid LTR-RT deletion and extensive haplotype variations in *S. lepidophylla*, we do not have direct evidence to support the conclusion of “genome shuffling”. Thus we removed this conclusion.

	LTR length*	Age (MY)	S/I
S. lepidophylla	512 ^a	0.371 ^a	4.06 ^a
O. sativa	535 ^a	0.924 ^b	2.71 ^b
A. thaliana	445 ^a	0.938 ^b	1.80 ^c

Supplemental Figure 7. Comparison of solo-intact (S/I) ratio of LTR retrotransposons in three plant genomes. To remedy the LTR length effect in intra-element recombination, the 95 percentile of LTR length in *Selaginella lepidophylla* (90-1540 bp) was used to screen for LTR-RT families that fell within this size range in the rice and Arabidopsis genomes. The S/I ratio of each LTR-RT family was estimated and the distribution of each genome was plotted. The table shows the means of LTR length, LTR age, and S/I ratio of each genome using the length-filtered dataset. Different letters in each column indicate statistical significance by pairwise *t*-test ($P < 0.05$). Osat, *Oryza sativa*; Atha, *Arabidopsis thaliana*; Slep, *S. lepidophylla*.

Minor points

Line 116: Why would the concentration of LTRs explain why it is was possible to assemble such regions? Should not clusters of repeats be a reason for assembly breakage?

We deleted this sentence. The presence of haplotype specific LTRs facilitated accurate assembly of both haplotypes compared to regions that had few haplotype differences that were likely collapsed in the assembly. We say this elsewhere in the paper so we have removed this sentence to avoid confusion.

Reviewers' Comments:

Reviewer #3 writes his/her comment in the Remark to Editor section. (S)he thinks all concerns have been fully addressed and recommends acceptance.